# Perception and Educational Needs of Developmentally Supportive Care At-Home for Parents of Pre-Term Newborns

**DOI:** 10.3390/healthcare11121700

**Published:** 2023-06-09

**Authors:** Jeong Soon Kim, Hae Ran Kim

**Affiliations:** 1Department of Nursing, Chungwoon University, Hongseong 32244, Republic of Korea; jskim2022@chungwoon.ac.kr; 2Department of Nursing, College of Medicine, Chosun University, Gwangju 61452, Republic of Korea

**Keywords:** developmentally supportive care, parents, pre-term newborn, educational needs

## Abstract

After discharge from neonatal intensive care units (NICUs), the parents of pre-term newborns have to provide developmentally supportive care (DSC) to their children; thus, educational support for parents is essential. This study aimed to explore the lived experiences of parents providing DSC to their children born as pre-term newborns at home and to investigate their parenting-related needs. This study included 10 mothers who were identified through theoretical sampling. In-depth interviews were conducted for data collection. For data analysis, grounded theory was used according to Corbin and Strauss’s process. The mother’s perception and educational needs were characterized by the phenomena “Coexistence of familiarity and unfamiliarity” and “Desire for expert support”. Causal conditions include the “Incomplete education system” and “Gap between expectations and reality”. Contextual conditions include the “Fear of developmental disability” and “Lack of good evaluation criteria”. Intervening conditions include the “Difficulty in obtaining useful information”. Action/interaction strategies include the “Active information seeking” and “Continuing to provide DSC”. The consequences were the “Needs for professional educational support”. The core category was the “Parenting routine that continues without awareness” and “Hope to establish parenting system supported by multidisciplinary experts”. These results may provide the preliminary evidence base for suitable educational programs and for developing a social support system for parents.

## 1. Introduction

Due to increased age at childbirth and advances in assisted reproductive technology (ART), the rate of pre-term infant births has reached 8.5% in the Republic of Korea, 1.5 times higher than 10 years ago [1].

Pre-term infants have poor survival capacity, due to a lack of adequate intrauterine growth. As a result, they are admitted to the neonatal intensive care unit (NICU) for treatment upon birth. NICU care is essential to the basic survival of these infants, but it is also an environment replete with pain, noise, and excessive sensory stimulation. Hence, along with other congenital factors, NICU care contributes to the developmental delay of these infants [2,3]. In other words, pre-term infants can survive through NICU care, but their likelihood of having a healthy life may decline due to the addition of various triggers that induce developmental disabilities as they grow. Therefore, parents and health care staff dealing with pre-term infants must consider the possibility of problems pertinent to growth and development and must not cease in providing effective developmentally supportive care (DSC).

Since the 2000s, the DSC provided in NICUs has achieved tremendous improvements. Nursing intervention programs such as the Newborn Individualized Developmental Care and Assessment Program (NIDCAP) have been well established, and their effectiveness has been substantiated extensively in the literature [4,5,6]. In contrast, it is important to review the DSC provided at home after discharge. Most acute medical problems linked to survival are resolved by the time pre-term newborns are discharged from the hospital. However, compared to healthy infants, they require continuous management, including nutritional management, the prevention of respiratory complications, and the prevention of hearing impairment and language developmental delay. This requires parents to be well prepared through education about special parenting needs tailored to the characteristics of their child before discharge. To this end, NICU staff provide family education as part of the transitional care before discharge. However, parents feel that the current discharge education system is inadequate and wish to receive more systematic education and support [7,8,9,10].

Parents who experience the birth of a pre-term newborns unexpectedly face an array of negative emotions. During the NICU stay, they experience anxiety, feelings of loss, and attachment impairment [11], and as they care for their child at home after discharge, they experience symptoms similar to posttraumatic stress disorder (PTSD) such as flashbacks [12,13] and report higher parenting stress than parents of full-term infants [14,15]. Therefore, providing psychosocial support to the parents of pre-term newborns is essential.

To address these issues, various parental support programs have been implemented in many countries, and their effectiveness has been proven to some degree [16,17,18,19,20,21]. However, programs that support parents in providing DSC and parenting at home after discharge from the NICU are relatively lacking in the Republic of Korea compared to other countries. Furthermore, there are no official programs run by public organizations in communities other than the financial support provided by the government, such as financial support for the cost of inpatient care and diagnosis of disability [22,23]. In addition, the types of support provided vary widely across regions. Additionally, there is a lack of an official support system that provides professional support, such as practical parenting-related knowledge and emotional exchange [8,23]. Moreover, while there is consistent and diverse research on educational and support systems for parents during the NICU stay, research for parents who raise pre-term infants at home after discharge is still scarce, calling for more diverse research.

This study aims to explore the phenomenon and lived experiences of parents providing DSC to their infants or toddlers born as pre-term newborns at home and to investigate their parenting-related needs in-depth, ultimately establishing foundational data for developing educational programs for DSC.

## 2. Materials and Methods

### 2.1. Research Design

This qualitative study used the grounded theory methodology to explore the perceptions and educational needs pertaining to DSC at home among parents who raise infants or toddlers born as pre-term newborns.

### 2.2. Sampling and Participants

The inclusion criteria were parents raising an infant or toddler who had been born as a pre-term newborn for at least 3 months at home. Infants did not have a current diagnosis for a congenital disorder or severe disease. Ten participants were enrolled. The mean age was 33.7 years, and all of them were in their 30s. The participants stated that they obtained information about parenting from internet resources (online parenting community, online pre-term baby community, video exchange platform), parents or other people around them, and their labor and delivery hospital (Table 1).

### 2.3. Interview Process

Data were collected through in-depth interviews conducted with participants enrolled through a recruitment announcement posted on an online parenting community from May 2022 to October 2022. The interviews were conducted at the participant’s preferred time over Zoom. Consent for the recording of the interviews was obtained from the participants, and the following semi-structured, open questions were used in the interviews to allow the participants to express their opinions freely: “Do you know about developmentally supportive care at home?”; “What are the greatest challenges you face as you parent your child?”; “What contents do you wish to be included in the education about developmentally supportive care?” (Appendix A). We observed participants’ facial expressions and behaviors during the interviews and kept field notes. Each interview lasted approximately 45–60 min. Data were collected until theoretical saturation, where no new categories or themes emerged. The recordings were transcribed by a research assistant, and one of the authors compared them with the recordings and supplemented them as necessary. Any unclear areas during transcription were clarified or confirmed with the participant over the phone or via text messages.

### 2.4. Data Analysis

The data were analyzed through a process of open coding, axial coding, and selective coding as per the grounded theory methodology presented by Corbin and Strauss [24], two authors and one Doctor of Nursing with experience in qualitative research. For open coding, the three individually read the transcripts line by line to understand, name, and conceptualize the phenomena experienced by the participants. During this process, the three compared and reviewed the extracted codes and clustered, condensed, and categorized them. Any disagreements among the authors were resolved by reviewing the raw data as much as necessary.

During the axial coding, the codes extracted during open coding were connected to establish a paradigm model that included causal conditions, the context, the central phenomenon, intervening conditions, action/interaction strategies, and consequences. During the selective coding, an overarching category that can describe the relationships among all categories was identified.

### 2.5. Rigor

To ensure the neutrality of the findings, we strived to adhere to the guidelines for truth value, applicability, consistency, and neutrality presented by Guba and Lincoln [25]. To ensure the truth value, we used semi-structured questions, and the study sample was selected with reference to the inclusion criteria so that the phenomenon of interest was well expressed. Furthermore, the interview data were transcribed ad verbatim as much as possible, and the extracted codes were reconfirmed with the participants. To ensure the applicability, in-depth interviews were continued until data saturation occurred, and the data were analyzed with the help of advice from a clinical nurse. To ensure consistency, the data were collected and analyzed with the central question in mind, and the authors and one Doctor of Nursing analyzed the data concurrently and exchanged feedback to ensure the consistency of the analysis. To ensure neutrality, epoche was practiced during data collection and the analysis of results to minimize the authors’ prejudice while reflecting the participants’ experiences and opinions as much as possible.

### 2.6. Ethical Considerations

The study methodology and contents of the study were approved by the Korean National Institute for Bioethics Policy (KoNIBP) (approval no. P01-202205-01-019). The participants were adequately informed about the purpose and method of the study and the recording of the interviews. We obtained voluntary consent to participate in the study, and after explaining the freedom to withdraw from the study, coding of personal information, and confidentiality of the interview data, study consent was recorded, and a signed written consent was obtained.

## 3. Results

### 3.1. Category Analysis

Through an analysis of the interview data, 10 categories and 28 sub-categories of themes were identified for the perception and educational needs at home of parents who raise an infant or toddler born as a pre-term newborn (Table 2). Axial coding was performed to connect the categories and to create a paradigm model.

#### 3.1.1. Core Phenomenon: “Coexistence of Familiarity and Unfamiliarity”, “Desire for Expert Support”

“Coexistence of familiarity and unfamiliarity”

Parents were very unfamiliar with the term “developmentally supportive care.” However, most parents continue to strive to provide DSC, such as nutrition, rehabilitation, and disease management, to facilitate the healthy growth of their children as much as possible; it was just that they did not recognize that their parenting activities were DSC.


*“Developmentally supportive care? I’ve never heard of it… But I think I just held my baby as much as possible because he/she was a preemie. First, I just held him/her as much as possible, and next, I massaged him/her a lot. We did a lot of tummy time and the jumperoos and stuff… I just did everything I could with the wish that my child would be healthy.”*
(Participant 3)


*“I’ve never heard of developmentally supportive care so I’ve thought it’s something challenging, but it was something that I’ve been doing all the time. First, a lot of people in online communities talk about their child walking abnormally and their development being delayed and these are what people are actually worried about a lot of the times… So, I search for information to prevent this from happening… Wouldn’t all moms do this? That’s what parents do…”*
(Participant 4)

“Desire for expert support”

In addition to general parenting problems, the parents wanted help from reliable professionals regarding medical management and developmental problems pertinent to pre-term births. In particular, they could not trust the information obtained from online mom’s communities, so they wanted to depend on clear guidelines and reliable experts.


*“But no matter how much I write about this in the community, other moms only reply with things like ‘my baby was a vomiter too’… and no one clearly explained why they vomit. I think the lack of such detailed medical knowledge was what I was sad about,”*
(Participant 6)


*“There is no information about rehabilitation or development centers. Especially, I live in the countryside, and it’s especially more difficult for me to find such information. And it’s not like I can take my unstable baby to Seoul. So, we just search for data ourselves and just rely on those rules of thumb.”*
(Participant 10)

#### 3.1.2. Causal Conditions: “Incomplete Educational System”, “Gap between Expectations and Reality”

“Incomplete educational system”

NICU discharge education includes content about kangaroo care and mother–child interaction, but this education is initiated 2~3 weeks before discharge and is given limitedly during visit hours. Thus, the mothers could not receive adequate education regarding DSC. The topics of education were also limited, including breastfeeding, how to deal with emergencies, and complication management. Moreover, there was no system to connect parents to other educational support once they were discharged; thus, many parents were unable to immediately resolve problems that arose while taking care of their children at home. The parents of children without a severe developmental disability were ineligible for development-promoting programs, and private development centers could not be utilized owing to long wait hours and long physical distances.


*“I did receive discharge education, but it was just a video. Because COVID-19 was so serious, we couldn’t have an education session. I did have education about bottle-feeding my baby for about an hour before discharge. But then (when we went home,) my baby growled so much during the night and I didn’t hear about it at all. I thought that happened because I did something wrong and I didn’t know what to do…For some time I kept calling the NICU to ask…”*
(Participant 2)


*“Even if you post things on the community, that’s kind of limited too. All the information posted is relevant to Seoul and Gyeonggi regions, and we’re in the countryside, you know. Even when I ask the attending doctor, they say I don’t have to visit a rehabilitation center. But I still want to check at least once… It’d be great if there is any information about development-related centers in the region…”*
(Participant 2)

“Gap between expectations and reality”

During the NICU stay, parents prepare themselves for post-discharge care at home by searching the internet and reading parenting books. However, they soon realized that actual parenting at home is much different from the parenting knowledge and anything they had prepared for, and they suffered substantial mental stress.


*“In the book, it said you nurse every three hours but the baby would wake up after 1.5 h and things like that are such a surprise… I really followed the book and highlighted important things and studied the book like I studied for my college entrance exam…. I realize the book is wrong and I get frustrated… At first, I felt really betrayed and had a hard time psychologically. Even though I can laugh about it now…”*
(Participant 10)

#### 3.1.3. Context: “Fear of Developmental Disability”, “Lack of Appropriate Evaluation criteria”

“Fear of developmental disability”

Parents feared the possibility of their child developing a developmental disability after reading about various complications in online communities, even if their child currently has no diagnosis of developmental problems. In addition, because pre-term newborns have a high risk for potentially developing a developmental disability, the parents could not be at ease even if their child currently had no specific symptoms. In particular, their anxiety intensified as they read about cases of the poor prognosis of pre-term newborns. Further, as symptoms that frequently occur among pre-term newborns—such as tippy toes, stretching, and growling—may be early signs of a neurological disorder, parents’ fear of developmental disabilities increased when they observed these symptoms from their child.


*“When I look through the posts on the online community, I see a lot of stories similar to that of my child. I’m just wondering how my child will grow. I feel relieved when I read about kids that grew up well, but I can’t stop being nervous when I read about kids that did not do so well. So, I try not to read them but it’s really hard, you know. They say that even if the child doesn’t have symptoms right now, they can appear later on.”*
(Participant 7)


*“The neurosurgeon didn’t talk about anything in particular, and I went to the rehabilitation medicine but the doctor said my child has no problems with things like cognition because it’s caught up to the corrected age but gross motor is a little slow. It took a long time until my child was able to grab, stand up, and walk. And his/her foot arch is a little abnormal and I told the doctor and he/she said not to worry… I was still anxious and so I searched on the internet and had my child wear corrective shoes.”*
(Participant 4)

“Lack of appropriate evaluation criteria”

During the growth and developmental assessment, parents witnessed that the results varied depending on the corrected age and actual age, and different criteria were applied depending on the child’s situation, which confused their parenting direction. Furthermore, they were not given any clear information about the development assessment, which exacerbated their confusion. The parents wanted to learn about the clear guidelines or standards for developmental assessment and parenting.


*“I was kind of wondering too about whether I should use actual age or corrected age and all the doctors say different things… Some say I should use actual age for eating and some say I should use corrected age for amount of feed. (omitted) You know, that should be adjusted to the baby, but I really don’t know.”*
(Participant 8)


*“When I go for the infant and toddler check-up, the doctor just says to use the actual age, but the doctor that I’ve always seen in the outpatient clinic tells me to use the adjusted age for the solids. So, I don’t know what’s right.”*
(Participant 5)

#### 3.1.4. Action–Interaction Strategy: “Continuing to Provide Developmentally Supportive Care”, “Active Information Seeking”

“Continuing to provide developmentally supportive care”

DSC was established as a routine part of parenting provided by the parents to promote their child’s growth and development. In particular, the synergy between the characteristic dedication to parenting and responsibility of parenting featured by Korean parents cultivated a more active parenting attitude among these parents compared to the parents of full-term infants. Moreover, these parents responded more vigilantly to problems with the child’s health and development and were very actively providing DSC.


*“I think the most challenging part is the waiting. Well, I think you need to be patient, not compare your child with other children because your child has his/her own timing. (Omitted) There are several governmental supports for preemies. There is ‘nutrition plus system’. *** Foundation provides rehabilitation support. You learn about a lot of things when you go about and actively seek for information.”*
(Participant 1)


*“I use the *** app to see both corrected and actual age. The app shows the feeding amount for both corrected and actual age. My baby was born a month prematurely, so I used something in between. In addition to that, I am still trying hard to search for parenting methods that I can use.”*
(Participant 8)

“Active Information seeking”

Parents actively sought information needed for their parenting. The most common source they utilized was online communities for parents of preemies, and other common sources included parenting apps and YouTube videos. Additionally, parents actively utilized human-based resources, such as exchanging information with other parents of pre-term infants.


*“For parenting information, I most frequently visit online mom communities and preemie communities. When I want to know something specific, I search online. My child had a dimple, and there wasn’t much information on the internet. So I posted a question about dimple on the community, and people replied to my post with practical information, like there is this famous doctor and like observe for these symptoms. (omitted) The **** app has a feature that gives recommendations for development-appropriate plays like playing with balls and shapes, so I referred to that a lot.”*
(Participant 3)


*“In the online communities, people say things like it’s good to massage the neck and chest, but it wasn’t easy to find videos and stuff. So, I asked other moms and got a video. And I usually search on YouTube for videos.”*
(Participant 10)

#### 3.1.5. Intervening Conditions: “Difficulty of Obtaining Useful Information”

As they actively searched for parenting information, most parents felt that they hit limits in obtaining useful information that can be applied to their child in consideration of their child’s characteristics. Furthermore, most pieces of information were non-expert information written by parents’ experiences, so they were not completely convinced about utilizing the information.


*“Online communities offer an array of information, but one drawback is that sometimes, some moms post inaccurate information that is not filtered out, and other moms who see them blindly follow them. Also, sometimes they say I’m doing this but other people shouldn’t do it depending on their case and that kind of causes confusion… (omitted) For example, when people tell me that I shouldn’t use stuff like baby gym when my baby begins to roll over because it’s bad for the stretching and bone development, I search and find that it’s a recommended toy… I was confused as to who should I listen to and I began to lose trust in the online moms’ communities.”*
(Participant 2)

#### 3.1.6. Results: “Needs for Professional Educational Support”

Parents strived to promote their child’s growth and development but hit limits in terms of expertise, and they direly wished for systematic and practical support for their parenting. In particular, parents showed a high necessity for education and support from experts who can give clear guidelines for their parenting. More specifically, they most wanted to know about nutrition, such as nursing and solid foods, when their child was in infancy based on corrected age and about neurological development and rehabilitation when their child entered early childhood. Parents also needed education about medical diseases and sequelae management, as well as a social support system, such as financial benefits and insurance, and they wished for relevant educational infrastructure in their communities.


*“If an expert like the doctor… if people who have been in the field for a long time reply to my posts, then I think I’d be able to trust that much more than the online communities.”*
(Participant 8)


*“The limitation of online communities is that you don’t know who these people are and that instead of (getting help from) non-experts, I wish that there was a community tailored to my child’s characteristics to get help from.”*
(Participant 9)

### 3.2. Restructuring of the Storyline through Selective Coding

In this study, the DSC provided by parents at home for their infants or toddlers born as pre-term newborns was defined as the process of parenting based on active information seeking to promote the optimal growth and development of the child.

The core category identified based on this definition was “Continued parenting even when unaware and wished for a shared parenting system supported by multidisciplinary experts”. The parents were unfamiliar with the term “developmentally supportive care”, although it was simply the routine parenting activities they continuously engage in at home to provide care for their child. Because they know that their child has a high risk for developing growth and developmental disabilities due to multiple risk factors, they have always had anxiety to a certain degree and were actively engaging in parenting to resolve that anxiety. However, they felt that they hit limits because of non-expert information, unclear evaluation criteria, and limited availability of community resources; thus, they wished for professional educational infrastructure (Figure 1).

## 4. Discussion

In this study, the core category of perception and needs for DSC among parents who raise infants or toddlers born as pre-term newborns was “Continued parenting even when unaware and wish for a shared parenting system supported by multidisciplinary experts.” In their study on the experiences of mothers of pre-term newborns at home, Breivold et al. [10] reported that parents encounter an experience as if they are “seeing the light at the end of the tunnel,” and they wish to receive continued parenting support and consultation from nurses of health care facilities; this supports our findings. Further, our findings were consistent with the conclusion of Lyne et al. [12], that parents have no time and little support from other family members at home and thus need experts in the community through contact via radio or video platforms.

The key phenomena experienced by parents in terms of their perception and needs for DSC are “coexistence of unfamiliarity and familiarity” and “desire for expert support”. It show that parents do provide DSC but are not too confident and highlight the need to implement measures to promote DSC at home in the Republic of Korea. These results are in line with previous findings that parents who received consultation and education through the VIC consultations program oscillate between feeling confident in caring for the infant on their own and needing support from others [18]. Further, study findings that the parents of pre-term newborns feel a strong need for active intervention and support from experts in the communities after they are discharged from the hospital [9,10,12] support our results.

The casual conditions(“incomplete educational system”, “gap between expectations and reality”), and the contextual conditions (“fear of developmental disability”, “lack of appropriate evaluation criteria”), influenced these phenomena. These are contextually in line with the findings of Garti et al. [9], that mothers encounter negative parenting experiences due to parenting-related physical exhaustion, emotional flatness, and dissatisfaction with social support. In addition, our results were similar to those of a study on parenting experiences after discharge by Lyne [12] where, despite preparing (e.g., breastfeeding, infection control) to raise a preemie at home with the assistance of healthcare staff upon perceiving the need for education, parents had difficulty providing DSC, such as kangaroo care, due to their hectic everyday life after discharge. These results highlight the need to consider the situations in the Republic of Korea when providing interventions to promote DSC among parents.

Parents’ strategic actions were identified as “continuing to provide developmentally supportive care” and “active information seeking”. The strategy of “continuing to provide developmentally supportive care” differed from the findings of a study on the strategies employed by parents who raise pre-term newborns [26], where these parents reduce their responsibilities or roles and instead frequently depend on and ask for help from others. This discrepancy may be attributable to the fact that Korean culture emphasizes family over the individual, imposes strong responsibility on parents in their parenting roles, and considers parents’ sacrifice and commitment something natural. The strategy of “active information seeking” was consistent with previous findings that the parents of pre-term newborns searched for information and worked hard to discover their child’s potential and provide care for their child, including ensuring safety at home [10].

The intervening condition for parents’ use of their strategies was identified as “difficulty obtaining useful information”, which resulted in the consequent “need for professional educational support”. Many existing studies [9,10,12,27] showed that this is a shared need among most parents of pre-term infants as they raise their child at home after discharge from NICU, supporting our findings. In fact, considering that even parents who received video-based consultations at home feel that their parenting confidence grew and that they need continued expert support [18], expert-based education and parenting support programs should be developed in the Republic of Korea to enhance parents’ competence in providing DSC.

Finally, for the first time in the Republic of Korea, this study investigated the phenomenon of DSC among community-dwelling families of pre-term infants. Compared to other countries, the home nursing system for the families of premature babies is weak, including the short discharge education period, lack of home visit nursing system in medical institutions, and limited local public medical service [22]. This research is expected to be an important first step for DSC in the domestic situation. In addition, based on the research results, various nursing education programs for DSC can be used as basic data for development.

However, this study was conducted on subjects in some regions of the Republic of Korea, and the subjects were limited to mothers. Since this can provide biased information in the educational composition of parent education programs, it is suggested that future research including various subjects and regions be actively conducted to overcome these limitations.

## 5. Conclusions

Our results suggest the need to develop criteria to evaluate growth and development that parents can apply at home, foster a human network to prevent their psychosocial isolation, and implement various parenting educational support systems to enhance parental competency in providing DSC. In particular, education programs run by health care professionals warrant development to prevent problems resulting from relying on inaccurate information found on the internet. Moreover, a shared parenting system and educational infrastructure should be developed on web-based educational media to enable parents to keep abreast with the rapid advances in information technology and the commercialization of smartphones.

## Figures and Tables

**Figure 1 healthcare-11-01700-f001:**
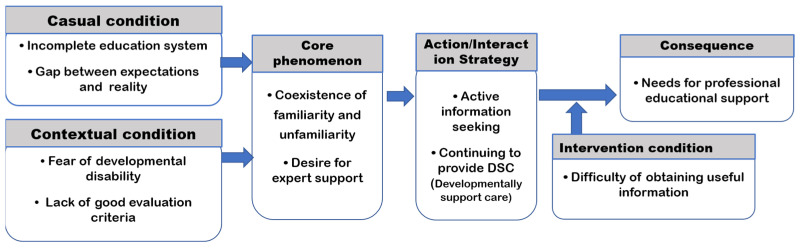
Paradigm of parent’s DSC Perception and Educational Needs.

**Table 1 healthcare-11-01700-t001:** Participants general characteristics.

Parents				Children			
Age (Years)	Gender	Job	Information Source *	Gestational Age (Weeks^+Day^)	Corrected Age (m)		Birth Weight (g)
32	Mother	No	B, C	1st: 28^+0^	48	Girl	1280
				2nd: 34^+6^	21	Boy	1580
32	Mother	No	B, E	27^+4^	14	Boy	1180
36	Mother	Yes	B	34^+5^	10	Boy	2000
37	Mother	No	B, C	26^+1^	31	Boy	840
33	Mother	No	A, B	1st: 31^+3^	36	Girl	1800
				2nd: 35^+2^	9	Girl	
36	Mother	No	B, C	28^+5^	14	Boy	1100
33	Mother	No	B	33^+6^	9	Girl	2030
34	Mother	No	B	35^+4^ (twin)	7	Boys	2110
							1910
32	Mother	No	B	1st: 26^+1^	31	Boy	670
				2nd: 35^+1^	14		1960
32	Mother	Yes	A, B	32^+2^ (twin)	25	Girls	1590
							1600

* Information source type: A (book), B (internet cafe, internet source), C (family or friends), D (public health center), E (NICU).

**Table 2 healthcare-11-01700-t002:** Paradigm, categories, and sub-categories of perception and educational needs of parents.

Paradigm Element	Categories	Sub-Categories
Casual condition	Incomplete education system	Inadequate discharge education Lack of post-discharge education network Limited resources at development center
	Gap between expectations and reality	Extensively studying parenting of pre-term newborns prior to being discharged
Contextual condition	Fear of developmental disability	Postnatal complications Indirect experience of negative prognosis Experiences of suggestive symptoms
	Lack of good evaluation criteria	Vague criteria for developmental assessment Conflict of expert opinions Criteria that change depending on the situation
	Coexistence of familiarity and unfamiliarity	Being unfamiliar with the term developmentally supportive care Continuing to provide parenting to promote development
	Desire for expert support	Lack of professional information about developmental rehabilitation Difficulty accessing a specialized health care facility Want an expert supportive program
Intervention condition	Difficulty of obtaining useful information	Difficulty sorting out the desired information from a flood of information Having no confidence in the information searched
Action & interaction strategy	Active information seeking	Using diverse internet resources Utilizing human resources to obtain information
	Continuing to provide DSC *	Continued effort to promote development Establishing one’s own parenting style
Consequence	Needs for professional educational support	Developmental rehabilitation support Nutrition support Health management support Governmental support and management Self-help communities Educational infrastructure for preemies

* DSC: developmentally supportive care.

## Data Availability

The data presented in this study are available from the corresponding author upon reasonable request. These data are not publicly available owing to privacy or ethical restrictions.

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
