# Peer review of "Perception and Educational Needs of Developmentally Supportive Care At-Home for Parents of Pre-Term Newborns"

_healthcare, 2023, doi:10.3390/healthcare11121700_

Round 1
Reviewer 1 Report
Thank you for giving me the opportunity to act as a reviewer for this manuscript.
When a child is discharged from a ICU, parents have to provide developmental support care to him/her, hence the educational support to them is crucial. In this way, the manuscript explores the experiences of parents providing DSC at home to their children that were born as high-risk neonates and delves into their parenting-related needs.
I found that the topic of the manuscript is interesting and relevant, and I think it is a timely and novelty research. In my opinion, it is an excellent work, taking into account the validity of the subject matter covered in the text and the compatibility of the study with the profile of the journal. Although this research study does not claim universal representation, the findings may be significant to educational practices.
The manuscript is written in a simple and clear way, both in its theoretical and methodological basis, and it is well structured. But the authors should carry out a new reading in order to correct some mistakes (for instance, ‘me/’ at line 287, or ‘. .’ at line 412).
The methodology is solid, but I must make two suggestions. In section 2.3. you can read that the interviews were semi-structured and that three questions were used (lines 101-105). I suppose that these questions were a starting point, and therefore more questions were asked, not only these three ones.
On the other hand, the number of researchers who participated in the analysis and the number of authors should be clarified: ‘the author’, lines 97, 109 and 135; ‘three authors’, lines 115, 116 and 118; but two authors below the manuscript title.
I consider that this manuscript does not require more modifications, maybe Corbin and Strauss should have made better observations, but no this reviewer.
Author Response
Point 1: The manuscript is written in a simple and clear way, both in its theoretical and methodological basis, and it is well structured. But the authors should carry out a new reading in order to correct some mistakes (for instance, ‘me/’ at line 287, or ‘. .’ at line 412)
Response 1: Thank you for your careful review. We have corrected these mistakes accordingly.
Point 2: The methodology is solid, but I must make two suggestions. In section 2.3. you can read that the interviews were semi-structured and that three questions were used (lines 101-105). I suppose that these questions were a starting point, and therefore more questions were asked, not only these three one
Response 2: Thank you for your suggestion. Additional supplements have been provided about semi-structured questions for Point 2.
Point 3: the number of researchers who participated in the analysis and the number of authors should be clarified: ‘the author’, lines 97, 109 and 135; ‘three authors’, lines 115, 116 and 118; but two authors below the manuscript title.
Response 3: Two research authors and one doctor of nursing participated in the study. We have modified the wording accordingly in the revised manuscript.

Reviewer 2 Report
I’m neither a native English writer, but this text can use some additional editing considerations.
The abstract is really difficult to understand, so that I highly recommend to reconsider the wording and text flow.
10 parents, or 10 pair of parents ? table 1 suggest that only mothers were recruited, or considered, while there is quite some emerging literature on the relevance to include fathers (or partners) in these studies. This is a major shortage, and should be further discussed. The same table suggest that the recruitment focus was on only former preterms ?
How ‘representative’ are the 10 mothers/parents included (age, profession etc) to the reference population ?
How do you define high risk infant birth ? I assume - based on the paper - that this risk refers to neurodevelopment impairment, so ? preterm ? very preterm ? congenital malformations ? others ? If so, this does not only refer to ‘poor’ intrauterine growth.
I highly recommend to add the semi-structured interview (questions) as a supplement to the paper.
How do you define developmental care ? it seems that this also includes infection prevention, respiratory care and/or nutritional support ? so broader than neurodevelopmental.
As mentioned by at least one of the parents, it seems that the infants were admitted during covid restrictions ? is this correct, and if so, has this affected the results as eg ‘bedside’ learning by doing and seeing, or parent/parent interactions has been much more limited
What is the ‘regular’ or standard follow up provided to high risks infants or former preterms after discharge ? and how do unit(s) provide information. The discussion suggest that these are ‘only poorly established in Korea’, but without any details. Based on the readings, it looks like parents only had very ‘general recommendations’ ?
I have read this paper with a background on clinical research in neonatology, be it with limited experience on qualitative research.
I’m neither a native English writer, but this text can use some additional editing considerations. The abstract is really difficult to understand, so that I highly recommend to reconsider the wording and text flow.
10 parents, or 10 pair of parents ? table 1 suggest that only mothers were recruited, or considered, while there is quite some emerging literature on the relevance to include fathers (or partners) in these studies. This is a major shortage, and should be further discussed. The same table suggest that the recruitment focus was on only former preterms ?
How ‘representative’ are the 10 mothers/parents included (age, profession etc) to the reference population ?
How do you define high risk infant birth ? I assume - based on the paper - that this risk refers to neurodevelopment impairment, so ? preterm ? very preterm ? congenital malformations ? others ? If so, this does not only refer to ‘poor’ intrauterine growth.
I highly recommend to add the semi-structured interview (questions) as a supplement to the paper.
How do you define developmental care ? it seems that this also includes infection prevention, respiratory care and/or nutritional support ? so broader than neurodevelopmental.
As mentioned by at least one of the parents, it seems that the infants were admitted during covid restrictions ? is this correct, and if so, has this affected the results as eg ‘bedside’ learning by doing and seeing, or parent/parent interactions has been much more limited
What is the ‘regular’ or standard follow up provided to high risks infants or former preterms after discharge ? and how do unit(s) provide information. The discussion suggest that these are ‘only poorly established in Korea’, but without any details. Based on the readings, it looks like parents only had very ‘general recommendations’ ?
Round 2
Reviewer 2 Report
no additional comments